# IS END-TO-END LEARNING ENOUGH FOR FITNESS ACTIVITY RECOGNITION?

## ABSTRACT

End-to-end learning has taken hold of many computer vision tasks, in particular, related to still images, with task-specific optimization yielding very strong performance. Nevertheless, human-centric action recognition is still largely dominated by hand-crafted pipelines, and only individual components are replaced by neural networks that typically operate on individual frames. As a testbed to study the relevance of such pipelines, we present a new fully annotated video dataset of fitness activities. Any recognition capabilities in this domain are almost exclusively a function of human poses and their temporal dynamics, so pose-based solutions should perform well. We show that, with this labelled data, end-to-end learning on raw pixels can compete with state-of-the-art action recognition pipelines based on pose estimation. We also show that end-to-end learning can support temporally fine-grained tasks such as real-time repetition counting.

## 1 INTRODUCTION

Action recognition in videos has slowly been transitioning to real-world applications following extensive advancements in feature representation and deep learning-based architectures. In many applications, models need to extract detailed information of the underlying spatio-temporal dynamics. Towards this, end-to-end learning has recently had a lot of success on generic action recognition datasets comprised of varied everyday activities (Carreira & Zisserman, 2017; Goyal et al., 2017; Materzynska et al., 2019). However, pose-based pipelines seem to remain the preferred solution when the task is strongly related to analyzing body motions (Bazarevsky et al., 2020; Shahroudy et al., 2016; Liu et al., 2020a;b; Yan et al., 2018), such as in the rapidly growing application domain of virtual fitness, where an AI system can be used to deliver real-time form feedback and count exercise repetitions.

In this paper, we present a new fitness action recognition dataset with granular intra-exercise labels and compare few-shot learning abilities of pose estimation-based pipelines with end-to-end learning from raw pixels. We also compare the influence of using different pre-training datasets on the chosen models and additionally train them for repetition counting.

Common approaches to generic video understanding based on end-to-end learning include combinations of 2D-CNNs for spatial feature extraction followed by an LSTM module for learning temporal dynamics (Donahue et al., 2017; Ng et al., 2015), directly learning spatio-temporal dynamics with a 3D-CNN (Ji et al., 2013), or combining a 3D-CNN with an LSTM (Molchanov et al., 2016). The temporal understanding can be further improved in a two-stream approach with a second CNN-based stream trained on optical flow (Carreira & Zisserman, 2017; Feichtenhofer et al., 2016; Simonyan & Zisserman, 2014). The large parameter space of 3D-CNNs can be prohibitive and efforts to reduce this include dual-pathway approaches to low/high frame-rate (Feichtenhofer et al., 2019) and resolution (Fan et al., 2019), temporally shifting frames in a 2D-CNN (Lin et al., 2019), and non-uniformly aggregating features temporally (Li et al., 2020). Using a multi-task approach, an end-to-end model jointly trained for pose estimation and subsequent action classification was shown to improve performance of individual components (Li et al., 2017) – but *pose information is still needed for training*.

Pose-based solutions for action recognition have two main stages: pose extraction and action classification. While bottom-up pose estimation approaches extract skeletons in one step (Cao et al., 2016;

| Datasets | Exercise Videos Dataset | NTU RGBD | FineGym | Jester | Smth-smth | Charades | Kinetics | MomentsIT |
|---|---|---|---|---|---|---|---|---|
| Focus on body motions | ✓ | ✓ | ✓ | ✓ | ✗ | ✗ | ✗ | ✗ |
| Fine-grained labels | ✓ | ✓ | ✓ | ✗ | ✓ | ✓ | ✗ | ✗ |
| Controlled environment | ✓ | ✓ | ✗ | ✓ | ✗ | ✗ | ✗ | ✗ |
| "In the wild" | ✓ | ✗ | ✓ | ✓ | ✓ | ✓ | ✓ | ✓ |
| Large-scale | ✗ | ✓ | ✓ | ✓ | ✓ | ✓ | ✓ | ✓ |

Table 1: Side-by-side comparison of the *Exercise Videos Dataset* (ours) versus common video datasets including NTU RGBD+D (Liu et al., 2020a), FineGym (Shao et al., 2020), Jester (Materzynska et al., 2019), Something-something Goyal et al. (2017), Charades (Sigurdsson et al., 2016), Kinetics (Kay et al., 2017) and Moments (Monfort et al., 2019) based on five criteria: a) focus on body motion, b) fine-grained label taxonomy (e.g. presence of intra-activity variations), c) controlled environment (e.g. fixed camera angle in a home environment), d) "in the wild" (as opposed to e.g. recorded in a lab), and e) dataset size sufficient for stand-alone pre-training.

Cheng et al., 2019; Newell & Deng, 2016; Geng et al., 2021), top-down methods split pose estimation into first localization and then pose extraction (Bazarevsky et al., 2020; Newell et al., 2016; Sun et al., 2019; Xiao et al., 2018). The classification stage is then optimized independently, with no end-to-end finetuning of the whole pipeline. Pose-based action classifiers typically use either hand-crafted features (Ofli et al., 2014; Wang et al., 2012; Vemulapalli et al., 2014) or, increasingly, deep learning-based modules. Recent approaches have employed CNNs (Ke et al., 2017; Kim & Reiter, 2017; Li et al., 2017), LSTMs (Liu et al., 2016; Zhu et al., 2016; Shahroudy et al., 2016; Zhang et al., 2017), Graph CNNs (Yan et al., 2018; Thakkar & Narayanan, 2019; Si et al., 2019), or 3D-CNNs on top of pose heatmaps (Duan et al., 2021).

In addition to an appropriate model architecture, a dataset with a fine-grained action taxonomy is crucial to learning robust action representations. Existing RGB-based video datasets such as Kinetics Kay et al. (2017), Moments in Time Monfort et al. (2019) and Sports-1M Karpathy et al. (2014) are based on a high-level taxonomy and further, possess correlated scene-action pairings resulting in pronounced representation bias Choi et al. (2019); Li et al. (2018). These concerns can be mitigated through crowd-sourced collections of predefined labels where the same action can be collected from multiple workers such as in the Something-Something Goyal et al. (2017), and Charades Sigurdsson et al. (2016) datasets. However, the "everyday general human actions" within these datasets are loosely specified and left to the worker's interpretation resulting in a high inter-worker action variance. On the other hand, FineGym Shahroudy et al. (2016) focuses on specific fine-grained body motions but includes variability in camera position resulting in lower overall action salience. In contrast, gesture recognition datasets such as Jester Materzynska et al. (2019) control camera and worker positioning and additionally, constrain human motion to appropriately specified hand gestures. A similarly constrained dataset for exact human body movement, that also controls camera motion, does not exist and we believe home fitness is the perfect domain in which to create one as workers can be instructed to move in very specific ways to perform exercises.

Pose-specific datasets contain an additional layer of annotated skeletal joints obtained either through annotation of scraped video datasets (either manually or using a pose estimation model Liu et al. (2020b)) or a sensor-derived approach in constrained lab settings Shahroudy et al. (2016); Liu et al. (2020a).

We present a new crowd-sourced benchmark dataset to fill a gap in the dataset landscape (see Table 1): videos of fitness exercises in a home setting are recorded in the wild providing challenging scene variety while also following a fine-grained label taxonomy. We compare end-to-end action classification models with state-of-the-art pose estimation-based action classifiers and show that the end-to-end approaches can outperform the pose estimation-based alternatives, if the end-to-end models are pre-trained on a large and granular labelled video corpus. We also show that the pose estimation models themselves can greatly benefit from pre-training on the large labelled dataset.

## 2 THE *Exercise Videos Dataset*[1] – A NEW BENCHMARK DATASET

Fitness activities are defined by a well-constrained set of body movements outside of which an individual risks injury or ineffectiveness. There is an opportunity for AI systems to detect mistakes and

---

[1]The name is a temporary placeholder due to double blind submission.

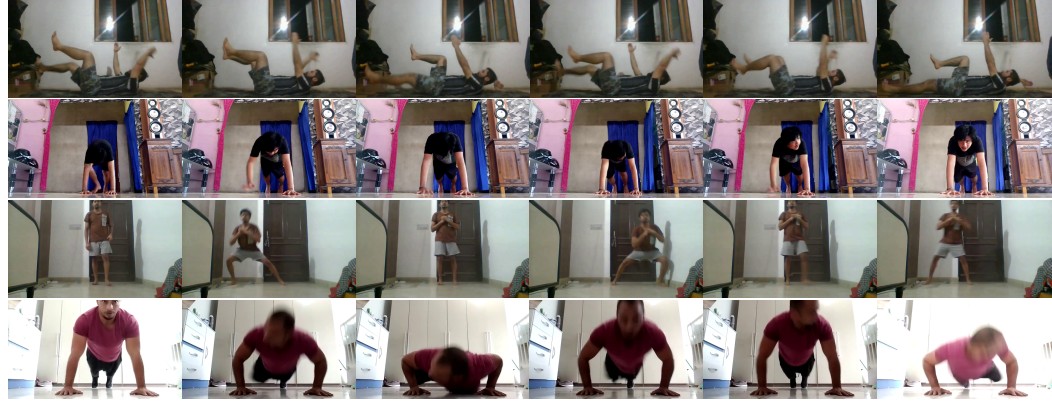

Figure 1: Samples from the Exercise Videos Dataset for all four exercises. From top to bottom: Dead bug, inchworm, alternating lateral lunges, spiderman pushups. Best viewed on a screen.

|  | Train | Validation | Test | Overall |
|---|---|---|---|---|
| Number of videos | 4000 | 711 | 800 | 5511 |
| Number of unique workers | 129 | 20 | 165 | 314 |

Table 2: Dataset statistics: Number of videos and unique crowd workers in each split

provide real-time form-correcting feedback. To this end, we present the *Exercise Videos Dataset* comprised of granular video-level activity classes capturing subtle variations, including common mistakes. The dataset spans four fitness exercises recorded in a home environment by crowd workers:

- *Dead bug*: The user lies on the back with arms and legs raised and moves them back and forth asynchronously.
- *Inchworm*: From a standing position, the user touches the floor with both hands, walks them forwards, and then back again.
- *Alternating lateral lunges*: The user performs a lunge step in sideways direction, alternating in both directions.
- *Spiderman pushups*: A pushup variation where one leg is moving to touch knee and elbow.

Example frames from the dataset for each exercise can be seen in figure 1. Each exercise was recorded with deliberate variations such as increased pace or incorrect execution of different aspects of each exercise, some which are visible from a static frame (foot touching the floor), and others which are only apparent across multiple consecutive frames (being too fast or too slow). In total, a fine-grained taxonomy of 40 video-level classes is available to trigger direct feedback.

Each of the 40 classes contains between 130 and 140 videos, with each video lasting between 5 and 8 seconds. The dataset is split into train, validation and test sets with no worker overlap between them. All videos are provided in MP4 format at a frame rate of 30 fps. The dataset contains 5511 videos in total across all splits (see Table 2 for details on the data split). For few-shot experiments, we prepared different versions of the train splits, containing fewer examples per class. We release splits that contain 5, 10, 20, 50 and 100 samples per class.

In addition to delivering form feedback in real-time, another challenging task for fitness AI applications is repetition counting. It relies on precisely parsing the temporal extent of an action segment within an activity and, as such, benefits greatly from the availability of temporal annotations. To this end, in addition to providing video-level labels, we tagged a subset of each exercise within the *Exercise Videos Dataset* with frame-level classes, thus making it possible to benchmark models on repetition counting. More details will be provided in Section 3.6.

The data has been collected in the wild by individual crowd workers who performed the actions following instructions from an example video. To match the desired viewing angle of a phone placed on the floor (fitness app scenario), the workers recorded themselves using a camera at a

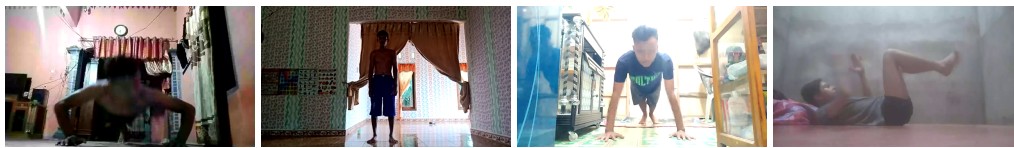

Figure 2: The videos in the dataset provide a wide range of lighting and scene settings. From left to right: cluttered background, textured background; high contrast, low contrast. Best viewed on a screen.

low position. All recorded videos were reviewed to confirm the execution was performed correctly. Because of the distributed nature of the data collection, the recorded samples show a large variety of scene settings, backgrounds and illumination (see figure 2). Each worker recorded videos for multiple action classes, so that the performed action cannot be learned by the visible video setting, but only by learning feature representations of the actual body motion.

The *Exercise Videos Dataset* has been collected for the purpose of discerning fine variations of exercise execution performed by a worker. In order to create the label taxonomy and recording instructions, the domain knowledge of several fitness experts had been consulted to collect a list of common mistakes and frequent variations of the individual exercises. Some examples of subtle variations are:

- *Dead bug*: A foot touches the floor; arms are not moving; the wrong leg is moving; execution is too fast
- *Inchworm*: Feet are too narrow or too wide; hands are too far from the body in the initial position; hands are stepping too far forward with each step
- *Alternating lateral lunges*: Bending the wrong leg; low range of motion; execution is too fast
- *Spiderman pushups*: Execution is too fast or too slow; leg movement is not in sync with pushup (three different error variations are labeled); pushup is too shallow

The full label taxonomy can be found in the supplementary materials. We plan to release the dataset under a non-commercial license, which permits non-profit research only.

## 3 EXPERIMENTS

All models were trained on subsets of the *Exercise Videos Dataset* training split, with 5, 10, 20, 50, and 100 samples per class, to evaluate few-shot behavior. Different initialization approaches were tested for each model, including training from scratch, starting from a pre-trained model and fine-tuning the final classification layer, all layers, or a subset of the layers. The approaches are described in more detail in section 3.4.

### 3.1 ARCHITECTURES

Three end-to-end and two pose estimation-based architectures are compared in our experiments. End-to-end architectures include I3D (Carreira & Zisserman, 2017), SI-EN (ours) and SI-BlazePose (ours). For the pose-based pipelines, we use *BlazePose* (Bazarevsky et al., 2020) to localize and extract human poses followed by one of two state-of-the-art graph-based classifiers: ST-GCN (Yan et al., 2018) and MS-G3D (Liu et al., 2020b). We selected *BlazePose* for the pose extraction part because it is optimized for real-time fitness applications and comparable to the end-to-end architectures in terms of FLOPs and model size (see section 3.1.3 for more details).

#### 3.1.1 END-TO-END

**I3D.** As an end-to-end baseline model for video action recognition, we used the 3D-CNN architecture, *I3D-RGB*, proposed in Carreira & Zisserman (2017).

**Strided-Inflated EfficientNet (SI-EN).** We present SI-EN, which uses EfficientNet-Lite4 (Tan & Le, 2019), a 2D-CNN, as a backbone, with a few modifications to some of the inverted residual

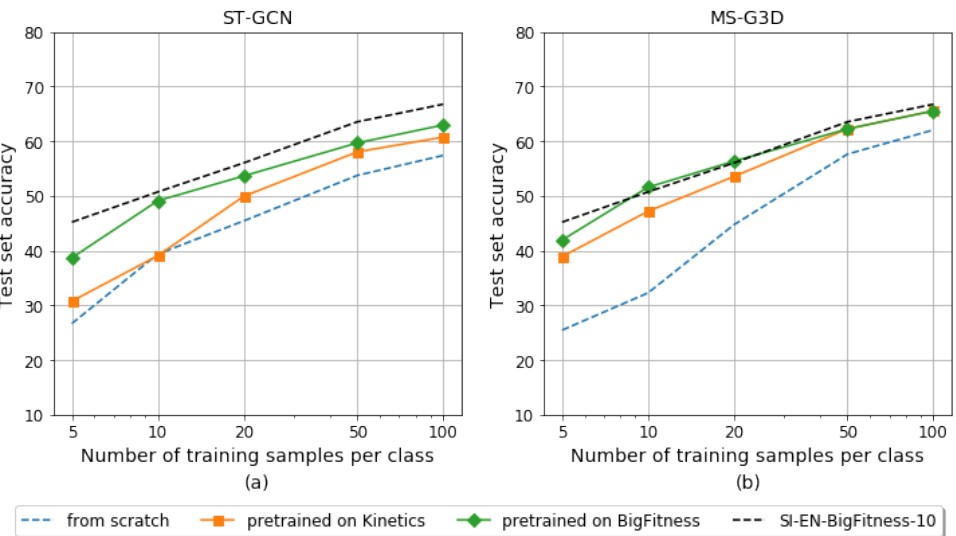

Figure 3: Effect of pre-training ST-GCN and MS-G3D on Kinetics and BigFitness.

blocks. Specifically, we inflate 8 of the blocks in the temporal dimension (blocks 3, 7, 11, 14, 17, 20, 23 and 25), using a temporal kernel of 3, effectively turning them into 3D convolutional modules taking inspiration from (Carreira & Zisserman, 2017). More precisely, it is only the first point-wise convolution in the inverted residual block that is inflated. Two of the inflated convolutions (blocks 7 and 14) are implemented with a stride of 2, enabling a lower footprint output of 4 fps from the 16 fps input stream.

**SI-BlazePose.** As a method to back-propagate through a pose feature bottleneck during an end-to-end classification task, we propose the following architecture which we call *SI-BlazePose*. It is based on the *BlazePose* model (Bazarevsky et al., 2020) using inflation to extend it in the temporal dimension. We inflate the last 8 point-wise convolutions with a temporal kernel of size 3, adding a temporal stride of 2 to the 2nd and 4th one. We freeze all layers before the first inflated layer. We use the full image as input, and resize it to $256 \times 256$ preserving the aspect ratio. We did not crop around the person as a first step, in contrast to what is done within *MediaPipe*[2]. Since the *Exercise Videos Dataset* is a classification dataset, we replace *BlazePose*'s body part regression head with a softmax layer.

### 3.1.2 POSE-BASED CLASSIFIERS

**ST-GCN.** Spatial-temporal graph convolution networks (ST-GCN) use graph convolutions across spatial joint connections and temporal connections from frame to frame (Yan et al., 2018). Following the original authors' approach, we included their suggested edge importance weighting method with a spatial partitioning strategy. As our results did not benefit from dropout regularization, we disabled it.

**MS-G3D.** Multi-scale graph convolutional networks (MS-G3D) adjust the node weighting in the graph for improved multi-scale aggregation and introduce skip connections to the graph for better modeling of spatio-temporal dependencies across longer distances.

As these models are able to work on generic graph layouts, we added support for the *BlazePose* layout by providing the adjacency matrix of the 33 graph nodes.

---

[2]https://google.github.io/mediapipe/

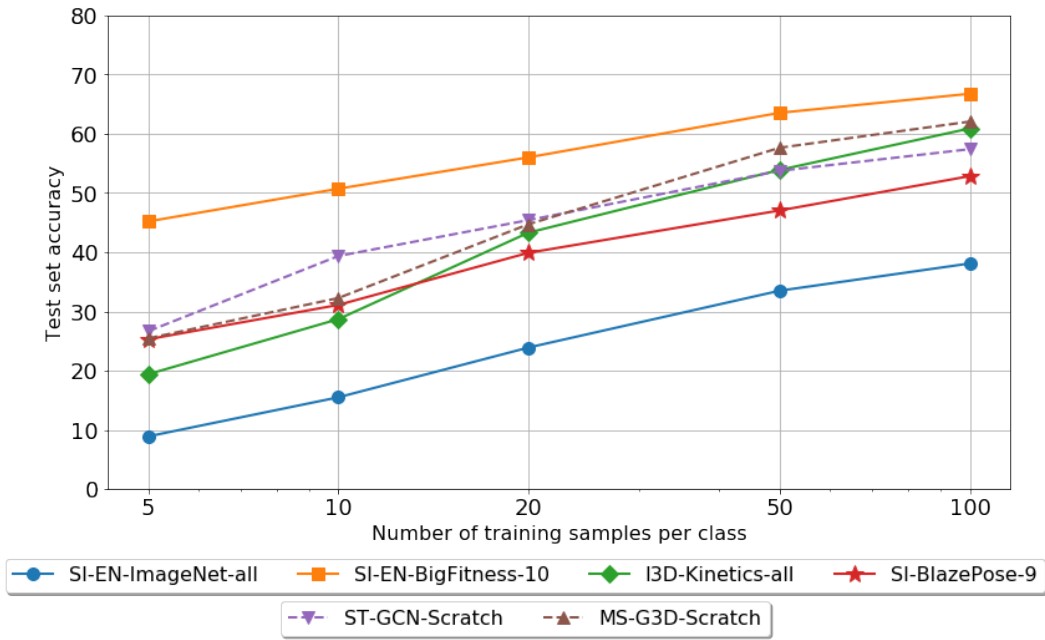

Figure 4: *Exercise Videos Dataset* top-1 accuracy of selected existing architectures, pretrained on various datasets. We report results using 5, 10, 20, 50, and 100 training samples per class. For each model, we use the following convention: {*architecture*}-{*pretraining dataset*}-{*optional: number of finetuned layers*}.

### 3.1.3 A NOTE ON COMPUTATIONAL EFFICIENCY

A pipeline based on pose-estimation typically consists of 3 components: a detection network producing rough person positions, a pose estimation network producing skeletons for each person (*BlazePose* in our case), and a classifier mapping a sequence of skeletons to an activity label (*STGCN* or *MSG3D* in our case). The first two components are image-based while the action classifier is video-based in the sense that it needs a sequence of skeletons. While the detection network can run fairly infrequently (at least, if the person is not moving their position much), the framerate at which the pose estimation component needs to run is determined by the temporal granularity required by the action classifier to obtain high accuracy. An end-to-end neural network on the other hand provides a variety of flexible ways to reduce computational footprint, e.g. by using temporally strided convolutions which reduces the framerate of subsequent layers and outputs. *SI-EN* specifically exploits this by introducing two 3D convolutions with a temporal stride of 2 early in the architecture. As a result, most of the *SI-EN* layers only need to run at 4fps rather than the 16fps input framerate, greatly reducing the computational footprint of our end-to-end solution. At an input framerate of 16, *SI-EN* only requires 4.0 GMACs/s, whereas running *BlazePose* alone (i.e. without counting localization and action classification) already amounts to 6.7 GMACs/s.

### 3.2 DATASETS USED FOR PRE-TRAINING

In addition to the dataset we are releasing along with this paper, we use a larger internal video dataset, which we refer to as *BigFitness*, for pre-training in some experiments. This dataset consists of around $300,000$ videos of fitness exercises with a fine-grained label taxonomy across $1,536$ classes that are disjoint from the data in the *Exercise Videos Dataset*.

In addition to this internal dataset, we also made use of Kinetics (Kay et al., 2017) and ImageNet (Russakovsky et al., 2015) for pre-training, as will be described in the results section. For more information about the relationship between the pre-training datasets used in our experiments and the *Exercise Videos Dataset*, please refer to Table 6 in the Appendix.

| | Number of samples per class: | 5 | 10 | 20 | 50 | 100 |
|---|---|---|---|---|---|---|
| End-to-end | SI-EN-ImageNet | 8.9 | 15.5 | 23.9 | 33.5 | 38.1 |
| | SI-BlazePose | 25.3 | 31.1 | 39.9 | 47.1 | 52.9 |
| | I3D-Kinetics-1 | 12.2 | 17.1 | 22.5 | 25.9 | 28.4 |
| | I3D-Kinetics-4 | 18.9 | 28.6 | 39.8 | 51.5 | 56.1 |
| | I3D-Kinetics-all | 19.4 | 28.7 | 43.3 | 53.9 | 60.9 |
| | SI-EN-BigFitness-1 | 38.1 | 44.4 | 49.5 | 56.0 | 58.9 |
| | SI-EN-BigFitness-10 | **45.2** | 50.7 | 56.0 | **63.5** | **66.8** |
| | SI-EN-BigFitness-all | 36.2 | 43.6 | 51.5 | 60.8 | 63.6 |
| Pose-based pipeline | ST-GCN-Scratch | 26.7 | 39.4 | 45.4 | 53.7 | 57.4 |
| | MS-G3D-Scratch | 25.5 | 32.3 | 44.7 | 57.6 | 62.1 |
| | ST-GCN-Kinetics | 30.8 | 39.1 | 49.9 | 58.0 | 60.7 |
| | MS-G3D-Kinetics | 38.9 | 47.2 | 53.5 | 62.2 | 65.6 |
| | ST-GCN-BigFitness | 38.7 | 49.1 | 53.6 | 59.7 | 63.0 |
| | MS-G3D-BigFitness | 41.9 | **51.6** | **56.3** | 62.2 | 65.5 |

Table 3: Results across all experiments. We report the test set accuracy in percentage on the *Exercise Videos Dataset*.

### 3.3 IMPLEMENTATION DETAILS

#### 3.3.1 END-TO-END

End-to-end models were trained on raw pixels from the *Exercise Videos Dataset* videos. The native resolution was down-scaled to a resolution of $256 \times 256$ pixels. To keep the original aspect ratio, frames were padded with black pixels to be in a square format before downscaling. Videos were subsampled to 16 fps which showed improved performance over the native 30 fps in preliminary experiments. For training, we took random crops of 63 frames from each video, which corresponds to roughly 4 second long video clips. 63 was chosen because of memory constraints. For evaluation, all frames of a video were passed to the model. As additional augmentation, we applied random color jittering to the 3 input channels. RGB values were scaled to the range from 0 to 1.

#### 3.3.2 POSE-BASED

To pre-train pose-based models in a way that is comparable to the end-to-end models, we extracted pose features from *BigFitness* using *BlazePose* (Bazarevsky et al., 2020) as provided by the *MediaPipe* library [3]. The same method was used to extract pose features to train on the *Exercise Videos Dataset*. In our experiments, we used all 33 joints and 3 input channels per joint: $x$ position, $y$ position and confidence score. The resulting pose sequences were created at 16 fps, because preliminary experiments showed better results than using the raw 30 fps (just like in the end-to-end experiments). For training, we took random crops of 90 consecutive poses. For evalutation, we passed in the full pose sequence of each sample. Following (Yan et al., 2018), we used simulated camera movement on top of keypoint coordinates as a data augmentation technique during training.

The Kinetics-Skeleton dataset (Yan et al., 2018), that we use for pre-training some of the models, uses the OpenPose (Cao et al., 2019) layout, which has fewer key points than the *BlazePose* layout (18 instead of 33). In our experiments, we mapped *BlazePose* keypoints to the OpenPose format with the neck position being defined as the center between the two shoulder joints.

### 3.4 RESULTS

The performance on the *Exercise Videos Dataset* across architectures is reported in Figure 4. For each model, we have tried multiple fine-tuning strategies (e.g. freezing all layers, fine-tuning a subset of the layers, fine-tuning the whole network). Figure 4 only reports the approach that worked the best for each model. Results obtained using the other strategies can be found in Table 3. Regarding pose-based baselines, to the best of our knowledge, there are no versions of *MS-G3D* and *ST-GCN*

---

[3]`https://mediapipe.dev/`. Note that we use the *GHUM Full* version of *BlazePose* in all our experiments.

|  | Temporal annotations schemes | pushup | dead bug | lateral lunges | inchworm |
|---|---|---|---|---|---|
| SI-EN | (1) within-repetition vs end-of-repetition | 25.9 | 39.3 | 33.3 | 109.0 |
|  | (2) within vs middle-of vs end-of-repetition | 17.1 | 22.3 | 13.4 | 49.2 |
|  | (3) first half vs second half | **4.6** | **7.2** | **2.2** | 21.5 |
| MSG3D | (1) within-repetition vs. end-of-repetition | 22.2 | 40.1 | 38.4 | 102.0 |
|  | (2) within vs middle-of vs end-of-repetition | 10.6 | 27.5 | 9.0 | 51.0 |
|  | (3) first half vs second half | 4.9 | 8.5 | 4.2 | **17.2** |
| STGCN | (1) within-repetition vs. end-of-repetition | 37.3 | 81.8 | 66.3 | 144.0 |
|  | (2) within vs middle-of vs end-of-repetition | 11.9 | 12.8 | 7.0 | 46.5 |
|  | (3) first half vs second half | 6.0 | 13.7 | 3.6 | 22.0 |

Table 4: Repetition counting results across all experiments (mean absolute percentage error).

pre-trained on the 33 joints returned by *BlazePose* and we therefore train the two graph CNNs from scratch in this experiment. We investigate the effect of pre-training *MSG-3D* and *ST-GCN* in the next section. Interesting findings from Figure 4 can be summarized as follows:

*Best performance is obtained by an end-to-end network. SI-EN-BigFitness-10* tops all other approaches with a significant margin, including pose-based solutions that use a graph CNN initialized from random weights. The gap with pose-based pipelines is higher when training data is scarce (45.2% vs 26.7% for *ST-GCN-Scratch* in the 5-shots case) but shrinks as more training samples are available (66.8% vs 62.1% for *MS-G3D-Scratch* when the full trainset is used).

*Pre-training on a large video dataset is key.* Unsurprisingly, the type of data used to pre-train each baseline plays an important role in downstream performance. Best results are obtained by the model that was pre-trained on *BigFitness*, which is by far the most granular pre-training dataset considered in this experiment. The exact same *SI-EN* architecture pre-trained on *ImageNet* performs poorly. The *Kinetics* baseline, *I3D*, is roughly on par with pose-based pipelines. On the other hand, the inflated pose 2D CNN, *SI-BlazePose-9*, obtains decent results when few samples are available but gets significantly outperformed as more samples are available.

MS-G3D *seems more prone to overfitting than* ST-GCN. While *MS-G3D* outperforms *ST-GCN* when more than 50 training samples are available, *ST-GCN* gets better results in the 5, 10 and 20-shot cases.

### 3.5 CLOSING THE GAP BETWEEN POSE-BASED AND END-TO-END APPROACHES

In this section, we investigate the effect of pre-training the graph CNN component of pose-based pipelines. Pre-training is performed with two datasets: *Kinetics* and *BigFitness*. Results can be found in Figure 3.

Figure 3 shows that, even for a pose-based pipeline, pre-training on a large video dataset can boost classification accuracy. While an accurate frame-level pose representation alone obtains decent results, the overall solution greatly benefits from pre-training on videos. This suggests that training data that provides some understanding of the temporal aspects of human body motions is highly beneficial, even for pose-based models. While pre-training on *Kinetics* produces good downstream performance, pre-training on a more granular dataset such as *BigFitness* works better overall. When it is pre-trained on *BigFitness*, the *MS-G3D*-based pipeline is on par with the end-to-end baseline, and the advantage that *ST-GCN* has over *MS-G3D* in the lower data regimes vanishes. Additional metrics (e.g. confusion matrices, f-measures) can be found in the supplementary material.

### 3.6 LEARNING TO COUNT

To explore a more temporally fine-grained recognition task, we also experiment with end-to-end repetition counting ("how many times has a given exercise been performed?"). This is a common task, in particular, in many fitness applications.

Repetition counting is an inherently temporal prediction task. To train the networks on this task, we temporally annotated a subset of videos with frame-by-frame labels describing which phase of the

exercise the subject is at any moment in time. We use the same frame-rates as before (16 fps input, 4 fps output) and annotated 100 videos for each of the exercises in the training set. We use the same train/test split as above for evaluation. We experiment with various temporal annotation schemes that can be turned into counts after training: (1) marking frames as within-repetition vs. end-of-repetition, (2) marking frames as within-repetition vs. end-of-repetition vs. middle-of-repetition, (3) using a different encoding of the 3-way annotations in (2), by marking frames as first-half of a repetition (between end-of-repetition and middle-of-repetition) vs. second-half of a repetition (between middle-of-repetition and end-of-repetition).

We train the models by treating these annotations as a simple temporal classification task. For training, we concatenate the videos within a mini-batch along the temporal axis rather than stacking videos on the batch-axis. Since annotation schemes (1) and (2) are highly imbalanced, we weight the classification cost by 0.2 for the under-represented class "within-repetition" during training. For *SI-EN*, we only train the 10 final layers (as for the best model above).

We turn temporal classifications into counts at inference time by incrementing the count when the end of a repetition is detected. For annotation schemes (1) and (2), we only increment the count if an end-of-repetition event is followed by at least one middle-of-repetition event to avoid over-counting. Table 4 shows the performance of the models in terms of mean absolute percentage error (MAE) (Runia et al., 2018; Levy & Wolf, 2015). It shows that accurate counting performance can be obtained from the relatively small number of annotated videos. While performance is comparable across models, interestingly, even in this setup, the end-to-end approach *SI-EN* performs roughly on par with or better than the other approaches in most cases. In fact, it shows the best performance in all exercises except for "Inchworm" which unlike the other exercises, has a much smaller number of repetitions per video and yields overall lower accuracy. Note that only *SI-EN* can make predictions on-line. Overall, while a deeper analysis and comparison with other counting approaches is beyond the scope of this paper, we find that it is possible to obtain very accurate repetition counts entirely end-to-end. We also find that accuracy depends strongly on how temporal annotations are represented during training.

## 4    CONCLUSION

In conclusion, our experiments show that end-to-end training on large-scale labeled video datasets without any form of frame-by-frame intermediate representation can compete with pose-based approaches, even in the context of fitness activity recognition where one could assume that an accurate pose representation is all you need. More importantly, regardless of the selected approach, pre-training on a large and granular video dataset is a key ingredient to achieving good downstream performance. In fact, our experiments show that good performance in action recognition tasks is mostly a function of dataset size and label granularity and less of the choice of model.

**Limitations and broader impact.** The introduced dataset subserves research on end-to-end reasoning about human activities using an RGB camera. It can be used to study and benchmark model architectures and to rethink workflows in the development of end-to-end neural networks. However, the dataset in its current size and form may contain biases. Training on this dataset alone may, for example, lead to models whose behaviors could depend on a subject's age, gender, ethnic background, etc. As such, the dataset as defined is suitable only for performing the research needs described above. In addition, model behavior will be a function of camera angle, lighting, and possibly other random aspects of the scene, camera, camera-angle and the subject interacting with the model. As for positive impact, research towards enabling quantitative assessment of health and fitness-related activities with just a camera can democratize access to and can greatly improve individuals' understanding of such activities and help unlock their benefits.

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

SUPPLEMENTARY MATERIAL

| Fitness exercises | Video-level classes | Frame-level classes |
|---|---|---|
| alternating lateral lunges | knee over toe | left leg bent |
| | low range of motion | right leg bent |
| | no obvious mistakes | end-of-repetition |
| | no stepping | |
| | not alternating | |
| | stepping foot pointing away | |
| | too fast | |
| | torso bent forward | |
| | torso bent sideways | |
| | wrong knee bent | |
| dead bug | foot touching the floor | middle-of-repetition |
| | moving opposite leg | end-of-repetition |
| | moving same side | |
| | not moving arms | |
| | opposite knee too bent or too close to chest | |
| | too fast | |
| inchworm | arms too narrow | plank pose |
| | arms too wide | end-of-repetition |
| | excessively short | |
| | feet too narrow | |
| | feet too wide | |
| | getting into position | |
| | getting into position - hands too far | |
| | good form | |
| | head up | |
| | hips too low | |
| | not far out enough | |
| | stepping too big | |
| | too fast | |
| spiderman pushups | arms too narrow | low pushup position |
| | arms too wide | end-of-repetition |
| | good form | |
| | no pushup | |
| | not alternating | |
| | not synced (down - leg in - up - leg out) | |
| | not synced (down - leg - up) | |
| | not synced (down - up - leg) | |
| | shallow | |
| | too fast | |
| | too slow | |

Table 5: Label taxonomy of the Exercise Videos Dataset

| *Exercise Videos Dataset* exercises | Closest *BigFitness* classes | Closest *Kinetics* classes |
|---|---|---|
| spiderman pushups  | pushups - sloppy
burpee - no upright position
burpee - no jump  | push up
crawling baby
headbanging  |
| dead bug  | bicycle crunches - small torso rotation
bicycle crunches - medium torso rotation
bicycle crunches - head down  | situp
knitting
unboxing  |
| alternating lateral lunges  | skaters - single jump (right to left)
grabbing an off-screen towel
skaters - slow  | lunge
side kick
squat  |
| inchworm  | burpee (no pushup) - stepping feet forward
burpee (no pushup) - stepping feet back
roll down  | dribbling basketball
deadlifting
push up  |

Table 6: Comparing dataset similarity: for each the *Exercise Videos Dataset* exercise (column 1), we compute a prototypical feature vector and show its 3 closest class centroids in feature space within *BigFitness* (column 2) and *Kinetics* (column 3). It shows that *BigFitness* has multiple labels that are conceptually similar, which is to be expected as it contains fitness actions with a disjoint, but also fine-grained, label taxonomy. More general action recognition datasets like Kinetics have some fitness actions, such as push up and lunge, which resemble the exercises from the *Exercise Videos Dataset*. However, because of the more coarse label taxonomy, the next nearest neighbors can be very different (such as labels "head banging" or "unboxing").

A​DDITIONAL RESULTS

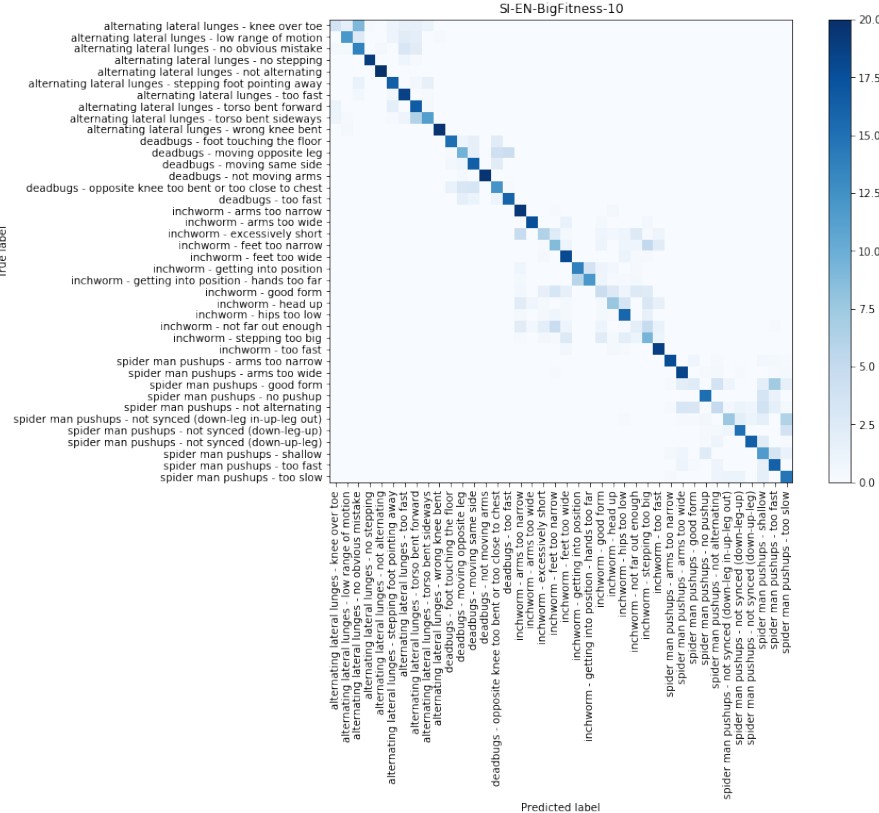

Figure 5: Confusion matrix for the SI-EN-BigFitness-10 model on the *FinestFitness* test set, averaged over five training runs.

| Exercise | SI-EN-BigFitness-10 | MS-G3D-BigFitness |
|---|---|---|
| alternating lateral lunges | 0.71 | **0.75** |
| deadbugs | 0.73 | **0.79** |
| inchworm | **0.55** | 0.48 |
| spider man pushups | **0.62** | 0.60 |

Table 7: Aggregated f-measures per exercise and model as an indicator of the intra-exercise performance, computed by averaging across all class-wise f-measures that belong to the same exercise. Each of the two examined models performs better on two out of the four exercises.

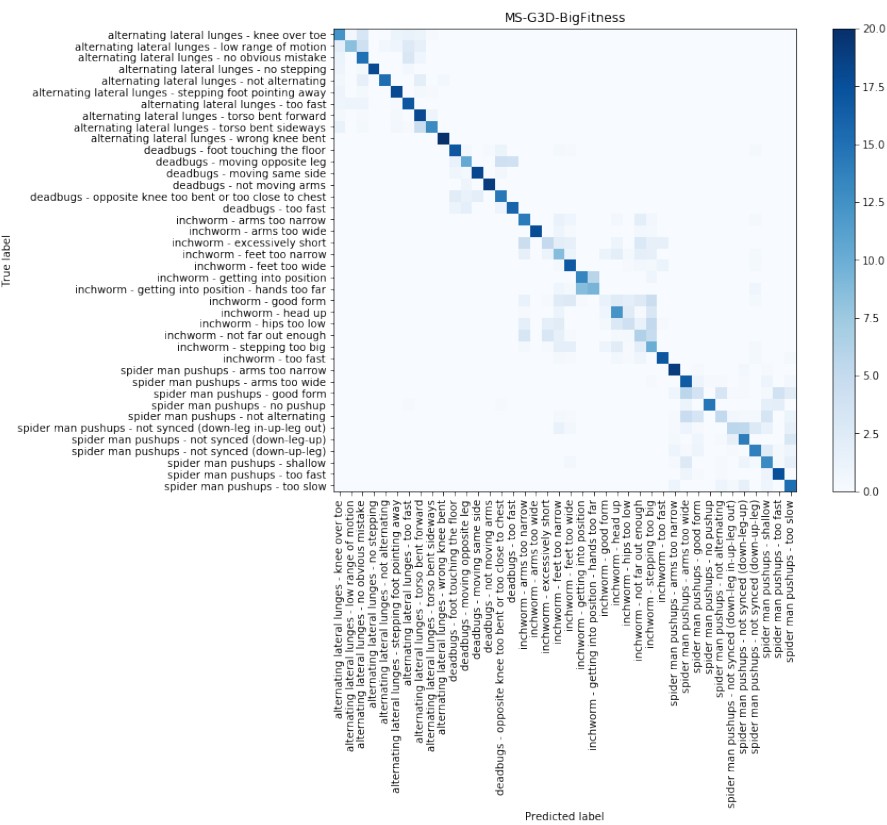

Figure 6: Confusion matrix for the MS-G3D-BigFitness model on the *FinestFitness* test set, averaged over five training runs.

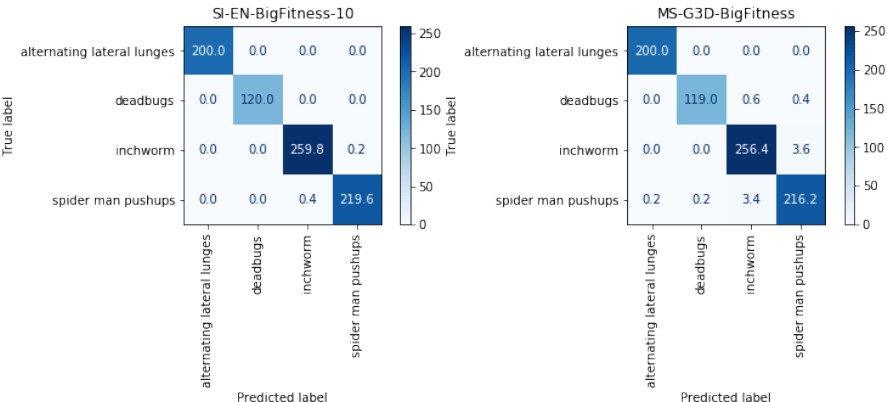

Figure 7: Confusion matrices from Figure 5 and Figure 6, aggregated into exercise-wide scores. Results have been obtained by summing up the scores for all classes belonging to the same exercise.

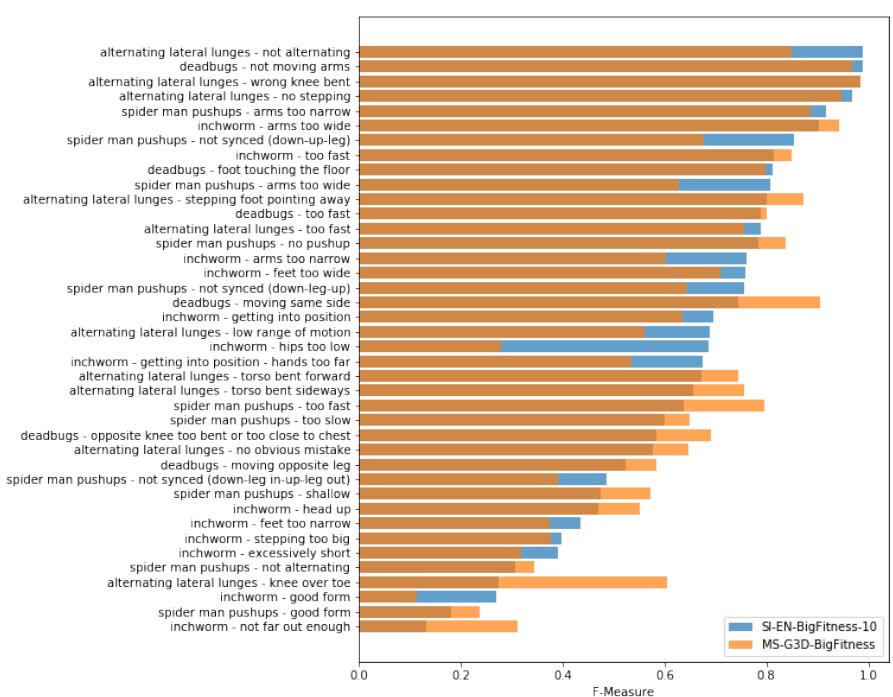

Figure 8: F-measures per class for SI-EN-BigFitness-10 and MS-G3D-BigFitness, obtained over 5 runs on the *FinestFitness* test set and sorted according to SI-EN-BigFitness-10 results.

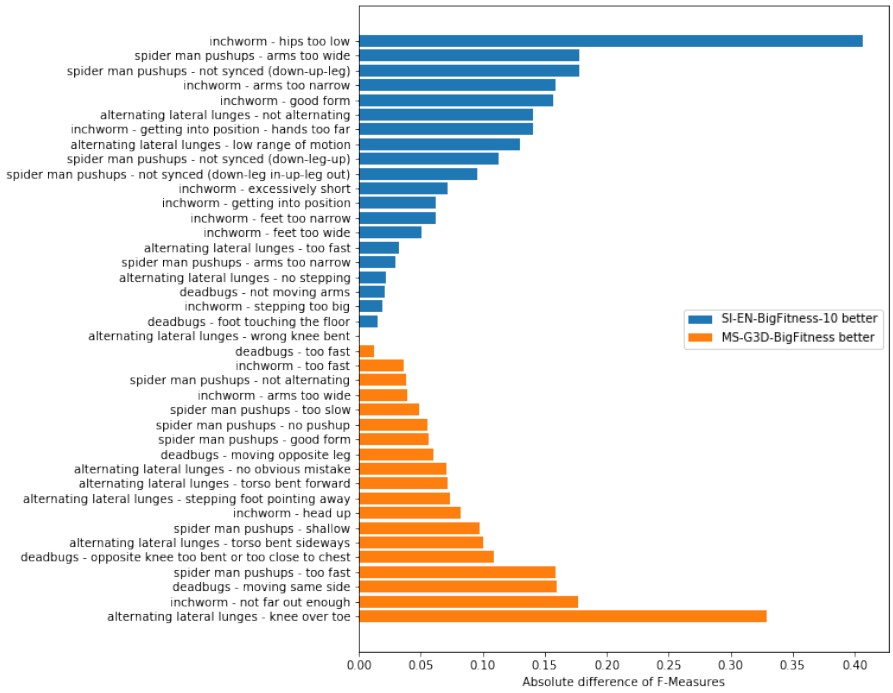

Figure 9: Absolute differences of f-measures for the two models from figure 8, sorted by decreasing performance of SI-EN-BigFitness-10.

