# OpenReview forum: "Is end-to-end learning enough for fitness activity recognition?"
_ICLR.cc/2023/Conference — Submitted to ICLR 2023_

### Official Review · Reviewer_GHiN · 2022-10-21

**Confidence:** 4
**Correctness:** 4
**Technical Novelty And Significance:** 1
**Empirical Novelty And Significance:** 1
**Recommendation:** 3

**Clarity, Quality, Novelty And Reproducibility:**

Clarity: the paper is relatively well written, but many important details are missing and captions are not sufficient to parse figure/tables.

Originality: extremely low. The dataset is similar to existing ones, no novel models are introduce, the conclusions from the experiments evaluation are very well known to the community.

Reproducibility: the large-scale dataset which is required to for the main experiments is not going to be released so the results will not be reproducible.

**Strength And Weaknesses:**

Strengths:

The paper is readable and is relatively easy to follow.

Experimental evaluation seems sound.

The conclusions that end-to-end training outperforms handcrafted features given enough data is accurate, though highly not novel.


Weaknesses:

The collected dataset does not seem to be valuable to the community since it is relatively small in scale and the difference to existing datasets is not significant. In particular, the authors argue that having a controlled environment (the only difference to FineGym), is a benefit. This argument requires a stronger justification, since in the other fields it has been consistently observed that the more diverse the dataset is the more robust the models trained on it are (e.g. COCO). The large scale version of the dataset would have been a significant contribution but the authors have no plans of releasing it.

Related work overview on action recognition is incomplete, essentially missing the entire body of work over the last 2 years. In particular, the transformer-based approaches are not discussed.

The main conclusion of the paper that pre-training deep network on large amounts of labeled data allows to outperform handcrafted features hasn't been novel since 2014. It is true that this has not been demonstrated for fitness activity recognition yet, but it has been for virtually every other task.

Experimental evaluation is incomplete. The choice of the architectures is not well justified (they vary from dated to extremely dated) and even some obvious variants are not reported (e.g. what if you pre-train I3D on your private data? would it outperform the inflated 2D convolutions?). In additions, the authors need to evaluate their approach on existing fitness recognition datasets to confirm that their conclusions hold (NTU, FineGym).

The presentation still needs improvement. The captions are not sufficient to parse the figures/tables, and some important details are missing (e.g. how were the 4 exercises selected? why is there no positive label for dead bug?).

**Summary Of The Paper:**

This paper contains two main contributions. Firstly, the authors have collected a new dataset for fitness activity recognition in a controlled environment (the setting of the videos is controlled by the authors). The dataset contains 5511 videos with 40 fine-grained activity labels. They then additionally collected a much larger, private dataset of 300 000 videos over 1536 classes and demonstrated that end-to-end activity recognition architectures (I3D or inflated 2D ConvNets) can perform on par or even outperform more traditional, pose-based approaches when pre-trained on a large amount of labeled data.

**Summary Of The Review:**

This paper is more of an empirical study which is only interesting in the narrow context of fitness activity recognition (no conclusions that would be interesting to the broader community). Moreover, the experiments are not reproducible since the authors are not planning to release the corresponding dataset. I find that this paper would be more suitable for a workshop on fitness activity recognition.

---

### Official Review · Reviewer_v9Hi · 2022-10-26

**Confidence:** 3
**Correctness:** 3
**Technical Novelty And Significance:** 2
**Empirical Novelty And Significance:** 2
**Recommendation:** 3

**Clarity, Quality, Novelty And Reproducibility:**

The paper presentation is clear, however the idea or motivation communication is umambiguous which make the novely (or claimed novelty) questionable. Meanwhile, if the BigFitness can not be released it is difficult to reproduce the performance claimed in the paper.

**Strength And Weaknesses:**

The strengths of the paper are:
+ Interesting perspective
+ good presentation
+ comprehensive experiments

The weaknesses of the paper are:
- Actually, I am not sure about the contributions of this paper, is it the dataset the authors created or the comparison experimental results the authors tried to show? The proposed and published dataset is Exercise Video Dataset, however it severed as a testing/evaluation dataset in the paper, while the BigFitness is the dataset experimented in the paper. I did not quite get the idea from the authors.
- The paper proposed that "end-to-end learning on raw pixels can compete with state-of-the-art action recognition pipelines based on pose estimation", however the testing dataset had the same domain as BigFitness, it is difficult to say the domain gap between the test/evaluation dataset or the pose estimation/pose oriented dataset provide the benefits.

**Summary Of The Paper:**

In this paper, the author asked and answered one question of action recognition, whether end to end learning is enough for the activity recognition. In order to answer this question, the author created a new labeled dataset, Exercise Video Dataset. With the labelled data, end-to-end learning on raw pixels can compete with state-of-the-art action recognition pipelines based on pose estimation.

**Summary Of The Review:**

Overall, I think the current paper did not provide a clear motivation and ideas to the community. In my opinion, it is more like a technical report for now while the experimental results are not convincing.

---

### Official Review · Reviewer_vBip · 2022-10-29

**Confidence:** 4
**Correctness:** 3
**Technical Novelty And Significance:** 1
**Empirical Novelty And Significance:** 1
**Recommendation:** 3

**Clarity, Quality, Novelty And Reproducibility:**

Novelty is somewhat limited. Many of these observations made by the paper have been already discovered by previous literature. There are no new insights obtained from the paper.

Data and source code are not available.

The Paper is well-written and easy to follow.



**Details Of Ethics Concerns:**

[ethics] How would the authors collect such datasets? Have the authors considered privacy issues (such as indoor home settings and faces)? Has IRB been sought and approved before data collection?

**Strength And Weaknesses:**

[Strength]
A new large human-centric dataset is proposed.
Paper is well-written

[Weakness]
The contributions of this paper are somewhat limited.
A new dataset in action recognition is proposed; however, there are no new insights that we can learn from. The empirical results from the paper align with many existing works of literature where these papers have already discovered and discussed about these observations.

The authors are strongly recommended to re-submit this paper to other venues which focus on dataset creation.

For improvements, I list the following suggestions:

[ethics] How would the authors collect such datasets? Have the authors considered privacy issues (such as indoor home settings and faces)? Has IRB been sought and approved before data collection?

[dataset availability] Would the authors make the dataset publicly available? If so, pls make such a statement in the paper.

[model development] Counting repeated actions in a video is interesting. The authors can specifically focus on this task and conduct more rigorous experiments and design new models.

[novelty] After training and comparing all the SOTA models on this newly curated dataset, what new conclusions can we draw from these? Any new insights? Any model behavioral analysis, such as attention?

**Summary Of The Paper:**

The paper proposed a new large human-centric action recogntiion dataset. From empirical results testing all the SOTA models on this dataset, the paper shows that end-to-end learning is more effective and it can support temporally fine-grained tasks such as real-time repetition counting.

**Summary Of The Review:**

Limited novelty, no new insights from the curated dataset, availability of the dataset is unknown,
A reject

---

### Official Review · Reviewer_zfp8 · 2022-11-01

**Confidence:** 5
**Clarity, Quality, Novelty And Reproducibility:** The quality, clarity and originality …
**Correctness:** 3
**Technical Novelty And Significance:** 2
**Empirical Novelty And Significance:** 3
**Recommendation:** 5

**Strength And Weaknesses:**

Strength:
1. The paper is written in a clear way and is easy to follow.
2. The effort of proposing a new public-available dataset is great.

Weakness
1. There are only four activities, which is really a small number compared to other datasets.
2. Datasets that focus on both human body movement and controlling camera motion have already existed. For example, PKU Multi-Modality Dataset is a large-scale multi-modalities action detection dataset containing 51 action categories in 3 camera views.
2. In table 3, why SI-EN-BigFitness-10 is superior to SI-EN-BigFitness-all? And in the meanwhile, I3D-Kinetics-4 is worse than I3D-Kinetics-all.
3. A minor suggestion: The author noted that end-to-end networks can reduce the runtime than that of pose-based networks. To make it a more straightforward comparison, can the author provide a running time comparison in the table?
4. In table 4, it’s better to add the average error among the four categories.
5. Downstream task is too simple. More evaluation tasks such as activity recognition and action localization are expected. Only conducting experiments on counting is not enough to claim the performance on action recognition tasks.


**Summary Of The Paper:**

This paper conducts comparison experiments showing that end-to-end training on large-scale labeled video datasets without any form of frame-by-frame intermediate representation can compete with pose-based approaches. In addition, pre-training on a large can help achieve good performance in action recognition tasks.

**Summary Of The Review:**

I have concerns about the uniqueness of the dataset and there are not enough evaluation tasks to support the claim. Addressing these issues will make this paper much stronger.

---

### Official Review · Reviewer_Pjes · 2022-11-01

**Confidence:** 4
**Correctness:** 2
**Technical Novelty And Significance:** 2
**Empirical Novelty And Significance:** 2
**Recommendation:** 3

**Clarity, Quality, Novelty And Reproducibility:**

The proposed fine-grained labeling seems to mix two separate tasks (i.e., action evaluation and action localization) but does not sufficiently refer to existing works in these two tasks (I will not provide them as there are many). No clear evaluation protocols provided will hinder future fair comparison.

**Details Of Ethics Concerns:**

The dataset might need subjects' 'to sign a consent for public usage.

**Strength And Weaknesses:**

Strength

1.	The varied scene and lighting settings are interesting. It would be good to provide an analysis of its effect, e.g., which exercises or classes can be highly affected by poor illumination conditions.

2.	End-to-end methods are compared with pose-based methods, where the former can occasionally be better than pose-based ones.


Weakness

1.	Given the poor performance of few-shot experiments, why not also report the fully supervised setting? There is no training from scratch results for end-to-end methods, while results of ST-GCN- and MS-G3D-Scratch can be based on a consistent number of joints with other pre-training-based results. Why some results in Table 4 are large than 100? You can report the ground-truth counting and the missed counting.

2.	The writing is not clear and some descriptions are not proper.
a.	“As a testbed to study the relevance of such pipelines”, what is the relevance?
b.	The dataset is collected in an indoor setting, referring to it as “In the wild” is not proper.
c.	What is GMACs?
d.	Sometimes in Table 3, Pose-based pipeline can be better than end-to-end methods. Why?

3.	Lack of discussion on some results, Table 3, the SI-EN-BigFitness-all underperformed than SI-EN-BigFitness-10. Why more pre-training data will lead to poor performance?

4.	There are no clear standard evaluation settings for few-shot learning and repetition counting. “Repetition counting is an inherently temporal prediction task”, there are many segmentation works that can be expanded as a counting task. The exact repetition counting setting is not clear and lacks comparison with existing works. What is the under-represented class ”within-repetition””?



**Summary Of The Paper:**

The paper collected a small-scale exercise video dataset with fine-grained labeling. Some baselines such as I3D, SI-EN, and GCN-based models (e.g., ST-GCN and MS-G3D) are provided for evaluation in the settings of few-shot learning. The baselines do not represent state-of-the-art models in their corresponding modalities, RGB video and Pose, respectively. Regarding the titled question, there are already plenty of results in the existing literature on datasets such as NTU RGB+D, which can draw the paper's conclusion.

**Summary Of The Review:**

Unclear writing, lack of justification for the proposed fine-grained labeling for the dataset, outdated baselines, and insufficient discussion on the experimental results make this work not ready for publication in its current form.

---

### Official Review · Reviewer_66o9 · 2022-11-02

**Confidence:** 3
**Correctness:** 3
**Technical Novelty And Significance:** 2
**Empirical Novelty And Significance:** 2
**Recommendation:** 3

**Clarity, Quality, Novelty And Reproducibility:**

As discussed in "Weaknesses", the current manuscript is not well-written and not reader-friendly.
I would like to suggest the authors could reorganize this manuscript. And it could provide clear research gaps, and highlight the significance of this work. I also suggest adding a related work section and a contribution list in the revision.

**Strength And Weaknesses:**

Strength:
1. It is good to study end-to-end training for fitness action recognition, and it is also good that it shows competitive performance with previous SoTA methods.
2. It could facilitate future research by establishing the fully-annotated action recognition video dataset.

Weaknesses:
1. The writing is not good, and the paper could be better organized.  As for the current manuscript, it is not easy for readers to get the research gaps, the significance of the achievements, and the main contributions of this work.
2. The proposed dataset seems to be very small, which makes the conclusion less convincing.


**Summary Of The Paper:**

This work targets studying end-to-end learning for fitness activity recognition.  A new fully annotated video dataset of fitness activities is established to evaluate different action recognition methods.  And they show that end-to-end learning could perform similarly to SoTA action recognition pipelines based on pose estimation methods.

**Summary Of The Review:**

This work seems to be an interesting study, but I did not recognize sufficient significant contributions.  As a result, I could suggest its acceptance.

---

### Official Review · Reviewer_ZAXq · 2022-11-03

**Confidence:** 4
**Correctness:** 4
**Technical Novelty And Significance:** 2
**Empirical Novelty And Significance:** 2
**Recommendation:** 3

**Clarity, Quality, Novelty And Reproducibility:**

The paper is generally well-written. The dataset is new. The experiments seem easy to reproduce.


**Strength And Weaknesses:**

Strength
1. The writing is clear and the proposed dataset is useful for fitness-related applications.

Weaknesses
1. Lack of novelty. This paper proposes a new dataset without additional innovation in terms of methods for this particular problem.
2. Missing references to recent fitness/repetitive action datasets [1*, 2*]. They should be compared and a discussion of how the proposed dataset is different would be useful.
3. Limitation due to fixed action classes. For the fitness action specifically, I think the authors should think about how the proposed dataset can help researchers scale to other types of fitness actions. Maybe class-agnostic is the way to go (like [1*])? The authors mention an internal 300k fitness dataset with 1.5k classes. Is the much larger dataset more useful in real-world applications?

[1*] Counting Out Time: Class Agnostic Video Repetition Counting in the Wild, CVPR 2020
[2*] TransRAC: Encoding Multi-scale Temporal Correlation with Transformers for Repetitive Action Counting, CVPR 2022

**Summary Of The Paper:**

This paper proposes a dataset for fitness action recognition. The dataset consists of 4 classes of fitness exercises with a total of 5511 videos. Fine-grained labels of 40 classes are provided to assess the quality of the actions. The videos are 5-8 seconds long so there is no need for temporal localization. The authors compare end-to-end models and pose-based models on the proposed dataset.

**Summary Of The Review:**

The dataset is small and there is no new method proposed. Some key related works are missing in the discussion.

---

### Official Review · Reviewer_2dWa · 2022-11-04

**Confidence:** 4
**Correctness:** 3
**Technical Novelty And Significance:** 1
**Empirical Novelty And Significance:** 1
**Recommendation:** 1

**Clarity, Quality, Novelty And Reproducibility:**

The paper is very poorly written and it’s very far from its final version. There are a lot of missing details as mentioned in the weakness section. There’s no novelty as such in terms of technical contribution. The results look reproducible to me based on the architectures used in the paper.

**Details Of Ethics Concerns:**

Since human subjects are used, has an IRB review been performed for the protection and welfare of human subjects?

**Strength And Weaknesses:**

Strengths

Since the dataset is collected in the wild, the dataset covers a wide range of lighting variations and scene settings. The proposed end-to-end architecture in general outperforms other architectures shown in the paper and is more computationally efficient than the pose-based architectures.

Weakness

1. The Paper lacks a lot of analysis.
    a ) Activity Selection - How these specific activities have been shortlisted? Do these four activities cover all types of variations we usually see in an Action Dataset?
    b) Architecture selection - How were the architectures selected for End-to-End training? The paper does not have an in-depth analysis to make a conclusion. There are multiple conventional 2D, 3D CNN, and transformer-based architectures that need to be tested on the dataset. There are also multiple architectures for pose-based that needs to be coveted for thorough analysis. Why was an efficient net selected as a backbone for SI-EN?

2. A table that compares the exact stats of the proposed dataset vs other action recognition datasets will help in the scale of the dataset and analysis of the variation of the dataset.

3. Missing details
  a) The number of epochs trained on pre-training datasets and the proposed dataset.
  b) The average duration of each activity in the dataset?
  c) Is the dataset trimmed or untrimmed?
  d) What is the original resolution of the video collected? How the quality of videos has been assessed?
  e) Is the dataset trained with 4 classes or 40 classes for the action classification task?
  f) Performance on the whole dataset training.
  g) In Table 3, what do 1, 4, 10, and all means? Why this setup is only for end-to-end and not for the pose-based pipelines?

4. Pre-training - How much overlap the internal training dataset has with the proposed dataset in terms of environment setup, viewpoint variation, and action length? These factors impact a lot. If the domain is the same, then that could be the result of better performance than Kinetics pre-trained.

5. The accuracy on the dataset is already between 60 - 70% with only 100 labels per class with baseline architectures. This doesn’t leave a lot of gaps in the improvement of the performance and questions how challenging the dataset actually is.

6. The comparison and a general comment of SI-EN outperforming other architecture for the Repetition counting tasks don’t seem correct to me. In Table 4, if we look into the 1st annotation scheme, MSG3D outperforms SI-EN in 2/4 of activities. For 2nd annotation scheme, STGCN outperforms SI-EN by a huge margin for all four activities (5-10%).

7. Missing citations - UCF101, HMDB51


**Summary Of The Paper:**

The work proposes a new video dataset on action fitness. The dataset covers a fine-grained taxonomy with a balanced distribution of videos per class. The paper proposes two new end-to-end architectures. An analysis of end-to-end vs pose-based architectures and their performance on the proposed dataset on two tasks - classification accuracy and repetition counting has been shown.

**Summary Of The Review:**

The paper lacks a detailed analysis both in terms of architecture setup and different viewpoints on which an analysis is shown on an action dataset (dataset stats, temporal analysis, more in-depth analysis of action classes, etc.).There’s no new problem that the proposed dataset is trying to solve and no new metric proposed in the paper for the proposed dataset. Due to these reasons, I tend to reject the paper.

---

### Official Review · Reviewer_a8ca · 2022-11-04

**Confidence:** 4
**Correctness:** 2
**Technical Novelty And Significance:** 2
**Empirical Novelty And Significance:** 2
**Recommendation:** 3

**Clarity, Quality, Novelty And Reproducibility:**

Clarity: well

Quality: normal

Novelty: not very

Reproducibility: not very

**Strength And Weaknesses:**

Strengths:

1. The proposed dataset contains fine-grained labels, including fine-grained video-level classes and frame-level classes.

2. The authors investigate models' performance in several aspects and evaluate them on an interesting repetition counting task.

3. The paper is clearly written and easy to follow.

Weaknesses:

1. The claims in this paper are all based on a very constrained scenery, the proposed Exercise Videos Dataset. I think the conclusions from existing well-known datasets are more convincing to me. Besides, the necessity of this new dataset is not well illustrated. Authors only claim that "everyday general human actions" are essential. The contribution does not meet the bar of ICLR.

2. Some conclusions are not novel.

    - Many state-of-the-art methods in action recognition opts the end-to-end training scheme. For example, DirecFormer\[1\] achieves the state-of-the-art performance on the Jester Dataset\[2\]. The paper of FineGym\[3\] compares a large range of methods with different input modalities. The methods with end-to-end training outperform the skeleton-based methods.

    - Pretraining is a well-known trick in both end-to-end networks\[4\] and pose-based action recognition\[5\].



\[1\] DirecFormer: A Directed Attention in Transformer Approach to Robust Action Recognition

\[2\] The Jester Dataset: A Large-scale Video Dataset of Human Gestures

\[3\] FineGym: A Hierarchical Video Dataset for Fine-grained Action Understanding

\[4\] Omni-sourced Webly-supervised Learning for Video Recognition

\[5\] Revisiting Skeleton-based Action Recognition


**Summary Of The Paper:**

This paper collects a new dataset for action recognition, which focuses on fitness activities. Based on this newly proposed dataset, the authors explore the influence of end-to-end training, pretrained dataset, and few-shot learning. This paper also compares these settings on a repetition counting task.

**Summary Of The Review:**

Despite I deeply appreciate the investigation in the new dataset, I find it difficult to justify the significance of the finding.

---

### Decision · Program_Chairs · 2023-01-20

**Decision:**

Reject

**Justification For Why Not Higher Score:**

Nine reviewers recommended the paper below the acceptance threshold or lower. There was no rebuttal.

**Justification For Why Not Lower Score:**

Lowest already.

**Metareview: Summary, Strengths And Weaknesses:**

Nine reviewers recommended the paper below the acceptance threshold or lower. There was no rebuttal.